# Structural Insights into Ca^2+^ Permeation through Orai Channels

**DOI:** 10.3390/cells10113062

**Published:** 2021-11-06

**Authors:** Yang Li, Xue Yang, Yuequan Shen

**Affiliations:** State Key Laboratory of Medicinal Chemical Biology and College of Life Sciences, Nankai University, 94 Weijin Road, Tianjin 300071, China; liyang1992@mail.nankai.edu.cn (Y.L.); yangxue@nankai.edu.cn (X.Y.)

**Keywords:** Orai, Ca^2+^ channel, STIM, SOCE, CRAC channel, Ca^2+^ permeation, gating, molecular mechanism

## Abstract

Orai channels belong to the calcium release-activated calcium (CRAC) channel family. Orai channels are responsible for the influx of extracellular Ca^2+^ that is triggered by Ca^2+^ depletion from the endoplasmic reticulum (ER); this function is essential for many types of non-excitable cells. Extensive structural and functional studies have advanced the knowledge of the molecular mechanism by which Orai channels are activated. However, the gating mechanism that allows Ca^2+^ permeation through Orai channels is less well explained. Here, we reviewed and summarized the existing structural studies of Orai channels. We detailed the structural features of Orai channels, described structural comparisons of their closed and open states, and finally proposed a “push–pull” model of Ca^2+^ permeation.

## 1. Introduction

Calcium ions (Ca^2+^) function as universal second messengers, playing an important role in almost every aspect of cellular life [1,2]. The endoplasmic reticulum (ER) is the main intracellular Ca^2+^ store. Therefore, maintaining Ca^2+^ homeostasis inside the ER is required for proper Ca^2+^ signaling [3]. Store-operated Ca^2+^ entry (SOCE) is one of the major pathways of extracellular Ca^2+^ influx [4]. Upon Ca^2+^ depletion from the ER, calcium release-activated calcium (CRAC) channels are activated and open to allow Ca^2+^ influx, resulting in an increase in cytosolic Ca^2+^ concentrations to generate sustained Ca^2+^ signals [5,6]. In this way, the entry of extracellular Ca^2+^ into cells through CRAC channels plays a major role in many cell types [2]. Consequently, mutations in CRAC channels cause various diseases, such as channelopathy and Stormorken syndrome [7,8].

The SOCE process requires two components: Ca^2+^ channels (Orais) and stromal interaction molecules (STIMs) [2,6,9]. Orais, as pore subunits of CRAC channels, are located in the plasma membrane (PM) [10,11,12]. STIMs are single-pass transmembrane proteins that are mainly localized to the ER membrane. STIMs sense the ER luminal Ca^2+^ concentration and then regulate the opening of Orais [13,14].

Here, we reviewed and summarized what is known about Orai dynamics and functions. We described the molecular features and gating of Orai channels. Furthermore, we discussed the molecular mechanism that allows Ca^2+^ permeation across Orai channels. The role of counterions in the flow of cations was emphasized.

## 2. Overview of Orai Channels

Three Orai genes (Orai1–3) and two STIM genes (STIM1 and STIM2) have been identified in mammals [5,6]. Orai1–3 are highly homologous to each other, as are STIM1 and STIM2 (sequence identity among Orai1–3 is 62%, and that between STIM1 and STIM2 is 54%). It is widely recognized that STIMs can directly activate Orais to generate CRAC currents [13,14]. Substantial structural and functional studies of Orais and STIMs have provided insights into the fundamental processes related to SOCE (Figure 1), which were previously reviewed [2,4,9]. Beginning with extracellular ligand stimulation, G proteins activate phospholipase C, and the latter decomposes phosphatidylinositol 4,5-bisphosphate (PIP2) in the PM into diacylglycerol (DAG) and inositol-1,4,5-trisphosphate (IP3) [15]. IP3 drains the ER Ca^2+^ store via Ca^2+^ release through IP3 receptors. Subsequently, ER-localized STIMs oligomerize and accumulate at ER–PM junctions. Oligomeric STIMs contact the negatively charged cytosolic face of the PM and directly interact with PM-localized Orai channels through their STIM1–Orai-activating region (SOAR) domains [16,17,18,19].

There are several notable features of activated Orai channels. The current through Orai channels is inwardly rectifying and voltage independent for tens of seconds. Similar to voltage-gated Ca^2+^ channels, activated CRAC channels show more than 1000-fold higher selectivity for Ca^2+^ ions than for Na^+^ or K^+^ ions [20]. Although both the negative membrane potential and the large Ca^2+^ concentration gradient across the PM ([Ca^2+^]_cyt_ is approximately 100–150 nM in the resting state, whereas [Ca^2+^]_ext_ is approximately 2 mM) drive Ca^2+^ influx, Orais have extremely low Ca^2+^ conductance and thus 100-times slower ion permeation than most Ca^2+^ channels [21]. When extracellular divalent cations are removed, the monovalent ions are able to permeate across the channel, and the unitary conductance increases from 7–25 fS in 2–110 mM Ca^2+^ concentration to approximately 200 fS in the absence of divalent cations [22,23,24]. Because of the high Ca^2+^ selectivity and slow permeation rate, Orai channels can produce sustained elevations in cytosolic Ca^2+^ concentration and prevent Ca^2+^ overload in cells.

STIM-activated Orai channels are essential for immune functions by activating immune response genes in T cells [8,25,26]. Loss–of–function mutants in both Orai channels and STIMs result in diseases such as severe combined immunodeficiency (SCID)-like disease. In addition, suppression of channel activity caused by loss–of–function mutations results in generalized muscle weakness, anhidrosis, ectodermal dysplasia, and tooth enamel defects [8,27,28]. In contrast to loss–of–function channels, Orai channels or STIMs with gain–of–function mutations are involved in tubular aggregate myopathy and Stormorken syndrome [29]. Drugs that regulate the channels have the potential to be used in treating related diseases, making the channel a clinically relevant target [2,30,31]. Consequently, a few CRAC channel blockers are now entering clinical trials [32,33].

## 3. Assembly of Orai Channels

Structural studies are good approaches to elucidate the molecular mechanism underlying Ca^2+^ permeation across unique Ca^2+^ Orai channels. The first Orai structural information came from the crystal structure of the *Drosophila melanogaster* Orai (dOrai) at a resolution of 3.35 Å (PDB code: 4HKR) [34]. The dOrai gene encodes 351 amino acids and shares 73% sequence identity with human Orai1 (hOrai1) in the transmembrane region. This dOrai structure, which is thought to be in a closed state, forms an expected hexamer arranged with three-fold symmetry (Figure 2a). Each subunit consists of four transmembrane helices (TM 1 to 4, Figure 2b) and a C-terminal helical cytosolic extension of TM4 (TM4e, Figure 2b). Six innermost TM1s are arranged practically perpendicular to the PM, forming a single narrow pore at the center. The TM2 and TM3 helices form a middle ring that encloses the pore formed by TM1, and this structure is surrounded by the outermost TM4 helices. Each TM4 helix is divided into TM4a and TM4b and bends at the conserved residue Pro288 (corresponding to P245 in hOrai1). Subsequently, the TM4e region extends into the cytosol; the coiled helices pack in pairs between every two subunits and surround the intracellular end of the channel. The TM4e helix is crucial for the binding of STIMs, undergoing conformational rearrangement when bound to STIMs.

The key pore-lining residues in TM1 with different properties divide the almost 55 Å long pore of the channel into four regions, from the extracellular region to the intracellular region (Figure 2c). A selectivity filter is formed by a glutamate ring (the E178 residue of each dOrai subunit corresponds to the Glu106 residue of hOrai1), which is negatively charged (the electrostatic surface potential map is depicted in Figure 2c). The E178N mutant has been reported to abolish Ca^2+^ permeation [35]. Then, a hydrophobic gate that is formed by three-layer hydrophobic residues (L167, F171 and V174) limits Ca^2+^ permeation. The V174A mutant dOrai channel is constitutively activated [34,36]. Finally, a positive charge blocker that consists of three layers of positively charged residues (R155, K159 and K163) further hinders Ca^2+^ permeation (Figure 2d). Orai channels with the R155S/K159S/K163S (SSS) mutations are closed [37]. The R91W mutation in the hOrai1 channel (corresponding to K163 in the dOrai channel) causes a severe combined immune deficiency-like disorder [28].

## 4. Activation of Orai Channels

To reveal the molecular mechanism underlying Orai channel activation by STIMs, an open structure of the Orai channel is absolutely necessary. Due to the difficulty in forming a stable complex between Orai and STIMs, the structural details of STIM-bound Orai channels are still unknown. A few gain–of–function Orai mutants used to explore channel activation were previously shown to provide a constitutively open pore. Recently, two crystal structures of dOrai mutants in the open state have been reported [38,39]. Together with numerous functional studies [40,41], the structural information of dOrai channels provided a glimpse of how the pore was rearranged upon activation and how Ca^2+^ permeates across Orai channels.

The H206A mutation of the dOrai channel (dOrai-H206A), which was shown to generate an activated channel with high selectivity for Ca^2+^ in the absence of STIMs, was obtained at a resolution of 6.7 Å using X-ray crystallography (dOrai-H206A_xtal_) [38] and later at a resolution of 3.3 Å using antibody-assisted cryo-electron microscopy (dOrai-H206A_EM_) [42]. Structural comparisons of open and closed dOrai structures revealed that the TM1 to TM3 regions are rearranged at the intracellular end and that TM4 undergoes drastic conformational changes (Figure 3a,b). The N-terminal regions of the innermost TM1 helices rotate to expand approximately 10 Å wide at the lower channel pore, and the hydrophobic region of the pore captures obvious dilation. From the side view, the antiparallel TM4e helices fully extend away from the TM1 to TM3 regions (Figure 3c). The H206A mutation (corresponding to the human H134A Orai1 mutation) is located in TM2. The substitution of the H206 residue with alanine may eliminate the steric hindrance that prevents TM1 from moving outward and possibly changes the intra-subunit interactions between TM1 and TM2, as well as between TM2 and TM3 (inset in Figure 3c); these phenomena could cause the rearrangement of the hydrophobic region. Hou and colleagues hypothesized that TM1 rotation within the pore is critical for pore widening and that the hydrophobic region acts as a ‘gate’ to prevent or permit ion conduction [38].

Our group reported the structure of another constitutively active dOrai mutant, P288L (dOrai-P288L), by both X-ray crystallography at a resolution of 4.5 Å and cryo-electron microscopy at an overall resolution of 5.7 Å [39]. Compared with the dOrai-H206A structure, the dOrai-P288L structure forms a similar hexameric assembly, but fewer dilated pores (Figure 3d,e). dOrai-P288L extends its TM4e region in each protomer away from each other into the cytosol. Owing to the replacement of P288 with leucine to fix the contact between TM4a and TM4b, the entire TM4 forms a fully extended helix in the dOrai-P288L channel. Additionally, the N-terminal regions of the six innermost TM1 helices twist away from the pore center axis (Figure 3f).

Within each protomer, the conformational transduction pathway starts at the peripheral TM4 helix and continues along the middle TM3 helix to the basic section of the N-terminal region of the TM1 helix (inset in Figure 3f). Mutations of hydrophobic residues in the hOrai1 channel, such as hOrai1-F257 in TM4, hOrai1-L261 in TM4 and hOrai1-F178A in TM3 (corresponding to F300, L304 and F250 in the dOrai channel, respectively), that interfere with the intra-subunit interactions between the TM3 and TM4 helices, cause a decrease in Ca^2+^ influx without attenuating the association of STIM1 with hOrai1 [39]. Furthermore, mutations that interfere with the intra-subunit interactions between TM3 and the basic section of TM1 in both open and closed Orai channels were also investigated. The L81A, E173A and H169A mutants of hOrai1 obviously attenuate or abolish extracellular Ca^2+^ influx [39,40]. These mutations were shown to have no effect on the STIM1–hOrai1 association. Thus, the revealed conformational transduction pathway (TM4b helix to TM3, then to TM1 helix basic section) is critical for Orai channel activation. It has been noted that this conformational transduction pathway was proposed in 2013 by Donald L. Gill’s group [43].

In the open state dOrai-P288L structure and the closed state dOrai structure, an anion binding pocket was observed around the basic section [34,39]. The quite compact basic region was proposed to provide electrostatic repulsion for preventing extracellular Ca^2+^ influx in the closed state. However, contradicting this proposal, mutations, such as R83A-K87A and R77A-K78A in the constitutively mutant hOrai1-P245L [39] and R155S/K159S/K163S in the open state dOrai-H206A [38], were introduced in the positively charged basic region to prevent electrostatic repulsion; these mutations greatly attenuate the influx of extracellular Ca^2+^. These results indicated that the basic region near the cytosolic face of the pore may play some unexpected but important roles in Ca^2+^ permeation.

## 5. Ca^2+^ Permeation

Orai channels are unique among all kinds of Ca^2+^ channels. Understanding the molecular details underlying how Ca^2+^ permeates through Orai pores will definitely advance the knowledge of ion conduction and chemical screening against related diseases. Both membrane potential and a 20,000-fold Ca^2+^ concentration gradient across the PM produce a substantial driving force for Ca^2+^ permeation through other Ca^2+^-conducting channels, but not through Orai channels. Ca^2+^ permeating through the pores of Orai channels must pass the selection filter, the unusual hydrophobic gate and the rarely seen positively charged blocker, as shown by structural investigation [34]. Yamashita and colleagues proposed a “rotation model” that overcomes these barriers during Ca^2+^ permeation; in this model, the pore helix is rotated to move the hydrophobic gate away from the central pore axis upon channel activation [44]. Later, Hou and colleagues proposed another “pore-dilation model” in which the hydrophobic gate and the positively charged blocker are dilated to allow Ca^2+^ permeation [38]. Although both models were elegantly designed and supported by many functional studies, they cannot reconcile the data that mutations in the positively charged blocker of constitutively active Orai mutant channels abolish Ca^2+^ permeation [45,46].

Recently, our group proposed an “anion-assisted Ca^2+^ permeation model”, which fits well with published functional results [39]. Here, we would like to rename this model, replacing its long name, “anion-assisted Ca^2+^ permeation model”, to a shorter name, “push–pull model” (Figure 4). Upon the Orai channel opening, the driving force from the membrane potential and Ca^2+^ gradient drives Ca^2+^ from the extracellular space into the pore. Inside the pore, the front Ca^2+^ will be pushed along the pore axis by the back Ca^2+^ due to cation–cation repulsion. Additionally, the positively charged blocker undergoes dilation and exposes positive charges toward the cytosol to aggregate anions. Thus, the Ca^2+^ inside the pore will be pulled by the anions located at the cytosolic end of the pore. The combination of the push force and the pull force presumably facilitates Ca^2+^ permeation through the hydrophobic gate.

## 6. Perspective

Structural studies of Orais, very unique Ca^2+^ channels, have provided an extensive understanding of channel assembly, ion selection, gating mechanisms and Ca^2+^ permeation. However, to date, only dOrai structure information has been obtained, and structural information on human Orai channels is lacking. Will the mechanism by which hOrai channels assemble be similar to that by which dOrai channels assemble? In earlier studies, Orai channels were proposed to be tetramers [47,48] or pentamers [49]. Until the crystal structure of the dOrai channel was published [34], Orai channels were generally accepted to be hexamers, and this was biochemically confirmed by several research groups [50,51]. Therefore, structural information on human Orai channels is expected.

## Figures and Tables

**Figure 1 cells-10-03062-f001:**
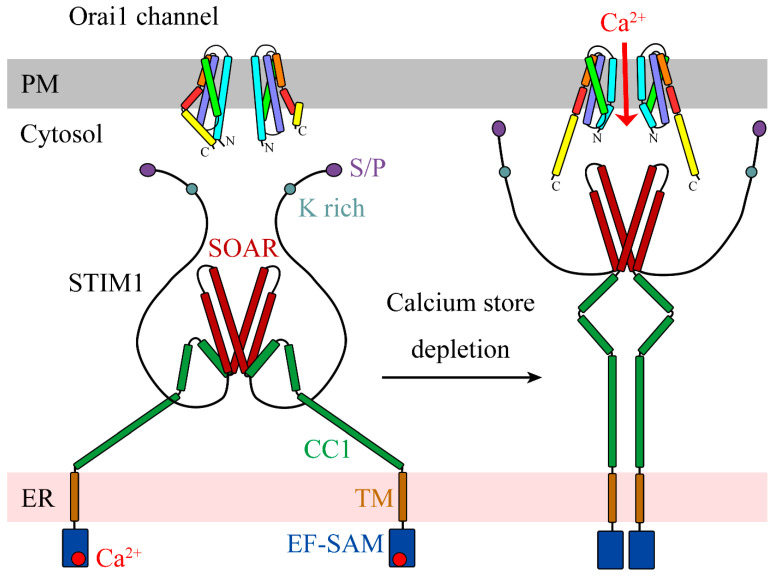
Diagram showing that STIM1 molecules activate Orai channels during SOCE. After ER Ca^2+^ store depletion, Ca^2+^ dissociates from the EF hand and the sterile alpha motif (EF-SAM) domain of STIM1. The N-terminus of STIM1 dimerizes to drive the conformational change of the CC1 region, eliciting the release of the SOAR domain. The SOAR then associates and activates the Orai channel on the PM. Finally, the open Orai1 channel allows extracellular Ca^2^^+^ to permeate.

**Figure 2 cells-10-03062-f002:**
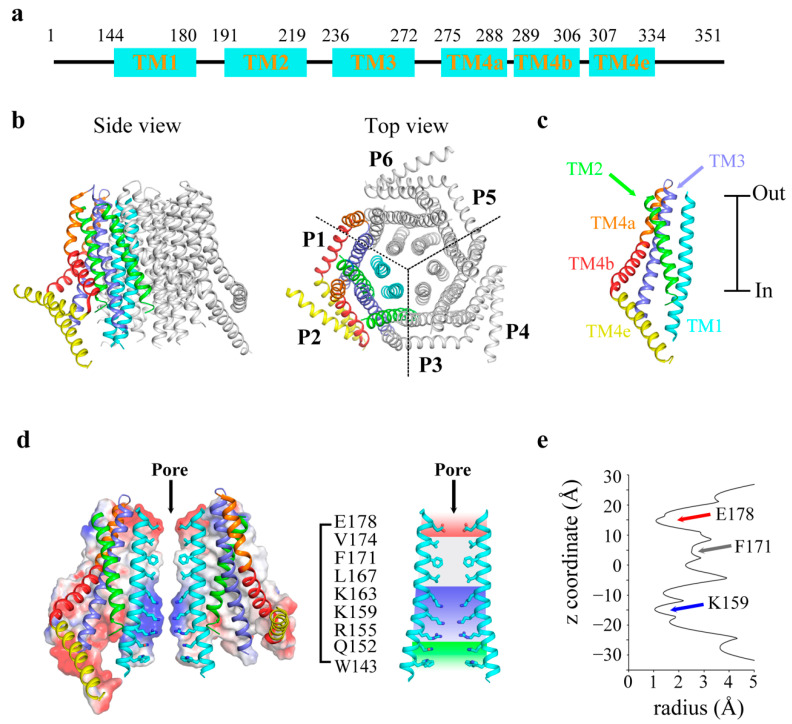
Structure of dOrai channel in a closed state. (**a**) Schematic diagram of the secondary structure labeled with the amino acid number of the dOrai channel. (**b**) Cartoon representation of the overall structure of the dOrai channel from the side and from the top. The dOrai channel shows a hexameric structure with three-fold symmetry. (**c**) Each monomer of the dOrai channel. TM1, TM2, TM3, TM4a, TM4b and TM4e are colored cyan, green, slate, orange, red and yellow, respectively. (**d**) The surface of two opposing subunits around the central pore are colored according to the electrostatic surface potential from −10 to +10 kT/e (red to blue). The proteins are shown as cartoons, and key residues lining the pore are shown as sticks (carbon, nitrogen and oxygen atoms are colored cyan, blue and red, respectively). Molecular details inside the pores are shown as the glutamate ring (the region colored in red) and hydrophobic (gray), basic (blue) and cytosolic (green) regions. (**e**) Pore radius profile of the dOrai channel along the z coordinate.

**Figure 3 cells-10-03062-f003:**
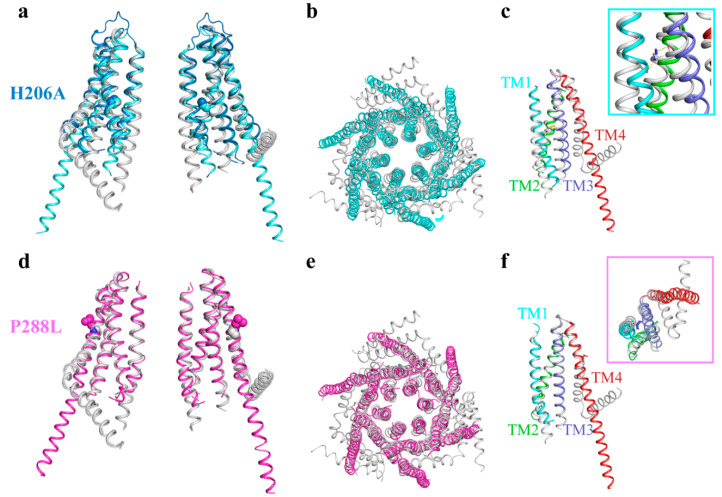
Structures of Orai channel mutants that are constitutively active. (**a**) Cartoon representation of two opposing subunits around the central pore from the structure of the dOrai-H206A mutant channel. Wild-type dOrai (PDB code: 4HKR), dOrai-H206A_EM_ (PDB code: 6BBF) and dOrai-h206A_xtal_ (PDB code: 7KR5) are superimposed together and colored gray, cyan and marine, respectively. Atoms of the H206 residue are shown as balls. (**b**) Top view of the overlay between the wild-type dOrai channel (gray) in a closed state and the dOrai-H206A_EM_ channel (cyan) in an open state. (**c**) Alignment of two monomers from the wild-type dOrai channel (gray) in a closed state and the dOrai-H206A_EM_ channel (multicolor) in an open state. (**d**) Cartoon representation of two opposing subunits from the structure of the dOrai-P288L mutant channel. The wild-type dOrai channel and the dOrai-P288L mutant (PDB code: 6KAI) are colored gray and magenta, respectively. (**e**) Top view of the overlay of the closed (gray) wild-type dOrai and the open (magenta) dOrai-P288L channels. (**f**) Alignment of two protomers from the closed (gray) wild-type dOrai and the open (multicolor) dOrai-P288L channel.

**Figure 4 cells-10-03062-f004:**
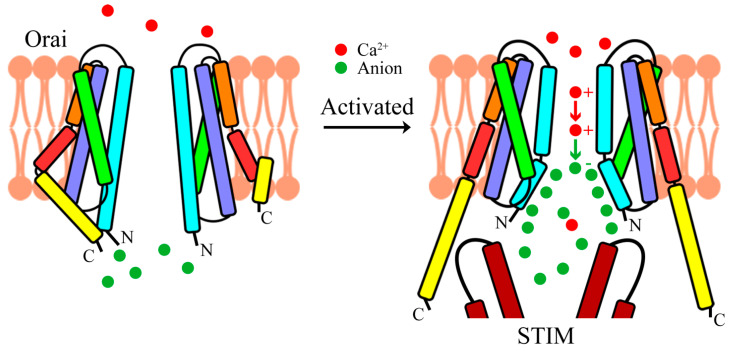
Proposed “push–pull model” of Ca^2+^ permeation. After STIM binds to the Orai channel, conformational changes in the Orai channel transduce along the pathway from the outermost TM4 helix to the innermost TM1 helix, inducing dilation at the cytosolic end of the TM1 helix. The Ca^2+^ ions inside the pore are pushed (denoted by the red arrow) by extracellular Ca^2+^ and pulled (denoted by the green arrow) by intracellular anions, thus passing through the pore of the Orai channel.

## Data Availability

The data presented in this study are available on request from the corresponding author.

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
