# Peer review of "Structural Insights into Ca2+ Permeation through Orai Channels"

_cells, 2021, doi:10.3390/cells10113062_

Round 1

Reviewer 1 Report

In this short review, the authors summarize the structural information collected nowadays about the mechanism of Orai channel activation, a calcium release-activated calcium channel. Particularly, they compare the channel conformation in its closed and open states obtained from the crystal structure of the drosophila wild type channel and of two mutated forms of this channel that remain constantly open. From these structures, they propose a functional model, in which channel activation (by mutation or by its endogenous regulator STIM-1) is due to the dilation of the cytosolic end of the pore, reducing the constrain linked to the hydrophobic region and removing the block of current at the basic region.

General comments

This review is relevant and understandable even for the reader not expert in protein structure or working on calcium channel mechanism of activation. I only have a few specific comments.

Specific points:

  • What does mean “the inside-out Ca2+ influx process” mentioned in the abstract ?
  • The causal relationship between the channel low conductance and high selectivity and the formation of Ca2+ hot spots around the channel (lines 69-71) is not obvious for me. Can the authors explain ?
  • For readers not familiar with the Orai channel structure, it would be useful to indicate the N and C-terminus ends of the monomers on the schemes in figures 1 and 4, as well as the total number of amino acids present in dOrai protein.
  • Line 114: apparently L167 (instead of L176), and line 116: K163 instead of K153.
  • References: end page numbers are missing in several places (for example; # 32, 37, 38, 40, 41, 42,44), or indicated twice (# 28).

Reviewer 2 Report

Li et al. have submitted a review manuscript entitled “Structural Insights into Ca2+ Permeation in Orai Channel” for publication in Cells. The manuscript is concisely written, adequately reflecting the authors’ background in the field. The following points should be considered to improve the article.

Major

Language generally needs to be improved, preferably by a native speaker. For example: articles and singular/plural forms, e.g. “The Orai channel belongs to the family of calcium release-activated calcium (CRAC) channels.

 “…cause noticeable arrhythmia symptoms, particularly the diseases that relate to the immune system”: Here and in other sections: please at least briefly name examples for better comprehensibility.

Further examples for major point:

“The SOCE process is made up of two components,”

“STIMs are a single-pass membrane protein”

“The Orai channel current is inwardly rectifying and voltage independence during tens of seconds.”

“Orais have extreme low Ca2+ conductance and accordingly 100-times slower of ion permeation than most Ca2+ channels”

“in local around the channel”

- and many more.

Round 2

Reviewer 2 Report

-